# What Aspects of Religion and Spirituality Affect the Physical Health of Cancer Patients? A Systematic Review

**DOI:** 10.3390/healthcare10081447

**Published:** 2022-08-02

**Authors:** David Almaraz, Jesús Saiz, Florentino Moreno Martín, Iván Sánchez-Iglesias, Antonio J. Molina, Tamara L. Goldsby

**Affiliations:** 1Department of Social, Work and Differential Psychology, Complutense University of Madrid, 28223 Madrid, Spain; dalmaraz@ucm.es (D.A.); jesus.saiz@psi.ucm.es (J.S.); fmoreno@psi.ucm.es (F.M.M.); 2Department of Psychobiology & Behavioral Sciences Methods, Complutense University of Madrid, 28223 Madrid, Spain; i.sanchez@psi.ucm.es; 3Department of Family Medicine and Public Health, University of California San Diego, La Jolla, CA 92093, USA; tgoldsby@health.ucsd.edu

**Keywords:** religion, spirituality, cancer, physical health, well-being

## Abstract

In recent years, the literature on the relationship between religion and spirituality (R/S) and the health of cancer patients has been flourishing. Although most studies focus on mental health, many study the physical health of these individuals. In order to summarize the findings of these studies, we reviewed the most recent research on this subject using the PubMed and PsycInfo databases. The objective of this systematic review was to recognize the primary R/S variables studied in research on physical health in cancer contexts. We found that spiritual well-being was the most-researched variable in studies of these characteristics, followed by R/S struggles and other variables such as religious coping; religious commitment or practice; or self-rated R/S. In general, R/S seems to have a positive association with the physical health of cancer patients, although the results are quite heterogeneous, and occasionally there are no relationships or the association is negative. Our results may assist in improving interventions that include spirituality in clinical settings as well as the development of holistic approaches, which may have a positive impact on the quality of life and well-being of cancer patients.

## 1. Introduction

Religion, the organized system of beliefs, practices, rituals, and symbols designed to facilitate closeness to the sacred or transcendent [1], has been part of human history since its inception. Spirituality may be considered intimately linked to religion, while also a unique and distinct concept. Koenig et al. [1], from a traditional point of view, define spirituality as the search for and discovery of the transcendent, as well as the relationship with the transcendent. Thus conceived, the two constructs are closely related. However, although conceptually, religious individuals may consider spirituality to be a central element of being religious, this need not be so from the point of view of predominantly spiritual people [2]. According to Schnell [2], spirituality may be conceptualized in two ways: a vertically transcendent spirituality that incorporates concepts of an eternal and supernatural God or higher power and a vertically transcendent spirituality that avoids reference to a supernatural reality but stresses the existence of an immanent absolute. It is noteworthy, therefore, that there are types of spirituality that do not require the existence of an organized system of beliefs or rituals. In this regard, Koenig et al. [1] highlight three additional ways of understanding spirituality apart from the traditional concept. The first of these may be considered a modern view of spirituality whereby spiritual people do not necessarily need to be religious and defined as “spiritual, but not religious”. A second manner of understanding spirituality would be a tautological view, which adds positive mental health and human values to the above. Finally, spirituality may be considered in terms of a modern clinical view, which would refer to it as another element that health care providers may consider in their approach to improving the well-being of patients. This can be seen, for example, in the relevance provided in certain health contexts, such as mental health, to religion and spirituality due to their significant implications in the prevalence, diagnosis, treatment, outcomes, and prevention of numerous disorders [3].

Koenig [4] argues that religion and spirituality (1) provide meaning and purpose during difficult life circumstances, which aid in psychological integration; (2) promote a positive view of the world that is optimistic and hopeful; (3) provide role models in the sacred writings that facilitate the acceptance of suffering; (4) give individuals a sense of indirect control over circumstances, reducing the need for personal control; and (5) offer a supportive community, both human and divine, to help reduce isolation and loneliness. Given these characteristics, religiosity/spirituality (R/S) may be considered a very useful tool to face difficult situations, such as an illness process. Thus, R/S is an aspect to be considered for health maintenance.

In recent years, religion and spirituality have begun to be studied in relation to health, with a substantial amount of literature now existing that documents the influence (or at least the association) of R/S on mental health [5,6,7,8], physical health [9,10], and even social health [11,12]. In general, R/S is not studied in a generic way. However, certain variables or dimensions are considered when describing this construct, such as attachment to God [13], trust in God [14], or spiritual well-being [15]. In fact, Hill and Pargament [16] propose that through these variables, which are conceptually more related to health, more can be discovered about the influence of religion and spirituality on physical and mental health. For this reason, numerous instruments have been developed to measure the different dimensions that make up R/S in health contexts [17].

As an aside, it is important to mention that this vast literature on religion, spirituality, and health is mainly composed of cross-sectional studies [18,19,20]. The difficulty of making causal inferences, the susceptibility to bias or the difficulty in interpreting the associations identified are some of the limitations that are typical of cross-sectional studies [21]. Therefore, it must be understood that many of the studies in this field have a series of problems that limit the possibility of making causal inferences. However, longitudinal studies, which are much more complex to carry out (especially in groups with serious diseases and therefore uncertain life expectancy), do allow for causal inferences.

Another limitation present in certain studies involves the small sample size. As stated by Hackshaw [22], studies with small samples may produce false-positive results or overestimate the magnitude of an association. These are problems that often appear in studies on religion, spirituality, and health, and that must be considered in our discussion regarding research in this field.

In any case, most of the research examining the relationship between R/S and health focuses on mental health, with a large number of studies showing the effects of R/S on this aspect of health [23,24,25]. As Koenig [26] discusses, it is to be expected that there are stronger relationships between mental health and R/S, since the latter involves psychosocial and behavioral aspects closer to mental health than to physical health.

However, as mentioned above, R/S has been studied in relation to physical health and has generally focused on the health of people with some type of physical pathology, such as respiratory disease [27], cardiovascular disease [28], or various types of chronic diseases [9,29]. Be that as it may, the relationships between R/S and physical health have been extensively documented [1,30,31,32].

It should also be made clear that since the constructs of physical and mental health have been shown to interact in numerous ways, it may be difficult to separate and identify the effective driving factors. Specific study designs, such as experimental RCTs or longitudinal epidemiologic design, may help in distinguishing the direction of effects. Authors such as Jones [33] have exposed this difficulty derived from the interaction between the physical and the mental. For this reason, numerous studies have been conducted involving mental health and physical health variables [34,35].

Nevertheless, part of this research has focused on the study of the relationship between R/S and physical health in cancer. In fact, although R/S may have general effects on health regardless of the disease, our study focuses on cancer, since focusing on this disease and its specific characteristics allows us to be more precise. In addition, oncologic disease is one of the leading causes of morbidity and mortality in the world [36] and results in significant physical, psychological, social, and spiritual impact on those who suffer from it. In general, the results obtained in these works are quite varied, usually as a result of the heterogeneity of the variables studied and the measures utilized, derived from the various existing conceptions of R/S [37,38].

In order to understand and summarize the heterogeneity of research results on the role of R/S in oncology patients, authors such as Jim et al. [39] conducted a series of meta-analyses that, again, appear to verify the diversity of results obtained, resulting, in some cases, contradictory results, as these findings go in different directions. These authors explored how the affective, cognitive, and behavioral aspects involving R/S affect the physical health of cancer patients. Generally, however, Jim et al. [39] found that R/S is associated with better patient-perceived physical health.

In view of this, we consider it relevant to inquire further into these results. Specifically, it is appropriate to examine the results found in research following the meta-analysis by Jim et al. [39] (which represents the latest review focused on the influence of R/S on cancer patients’ self-rated health), for several reasons.

In regard to the association of R/S and the physical health of oncology patients, research is necessary to understand the processes and mechanisms underlying this relationship [1,39,40]. If we can identify the different aspects of R/S that are associated with the physical health of these individuals and the psychosocial variables that may be involved in this association, we will be closer to identifying these mechanisms.

Moreover, through the identification of the variables included in these studies, it may be possible to determine potential moderating variables in these relationships [40], which may assist both in identifying mechanisms and explaining the manner in which relationships involved are strengthened or attenuated. This, in turn, may assist those who incorporate R/S as part of their therapeutic approach to guide their intervention. Likewise, following a more clinical view, identifying R/S variables and their impact on the health of cancer patients may assist in the development of treatment plans and/or holistic intervention in oncological contexts.

Therefore, in this study, we conducted a systematic review of the post-2015 literature with the aim of recognizing the main R/S variables examined by recent studies in relation to the physical health of cancer patients, as well as synthesizing the major results obtained in these studies. Thus, the question that guided our research was the following: how do various variables linked to religion and spirituality affect the physical health of cancer patients?

## 2. Materials and Methods

The quality standards of the PRISMA 2020 methodology [41] were followed to perform this systematic review. PRISMA 2020 is a reporting guide designed to address issues in the publication of systematic reviews, representing an important aid in the planning and conduct of these reviews by ensuring that all recommended information is captured [41]. The PRISMA 2020 checklist includes 27 items (some of which have sub-items) divided into seven sections or domains, related to each section of the review report. We used this guide for two reasons. First, this allowed the present authors to be cognizant of their own potential biases, thus minimizing the possible bias introduced in the conclusions and to guarantee the transparency of such a long and laborious process.

Second, the PRISMA 2020 guide was used as it represents an updated reporting guide that includes new recommendations and existing methodologies for conducting systematic reviews. Thus, it is a very complete tool that allows researchers to produce a transparent and accurate publication describing reasons the review was carried out, the procedure utilized by the researchers and what was discovered. After studying the PRISMA criteria and prior to the systematic review itself, a generic examination of the reviews prior to 2015 was performed in order to establish terminology and avoid duplication of concepts.

### 2.1. Search Strategy

A search was carried out in two specialized databases: PsycInfo and PubMed. The search occurred between February and March 2022, and included articles published between 2015 and March 2022.

The terms “religion”, “spirituality”, “cancer”, and “health” were used as search terms, linked to each other through the Boolean operator AND in order to objectify the search in any field, thus achieving a broader scope. Thus, the specific search string was “religion” AND “spirituality” AND “cancer” AND “health”.

### 2.2. Eligibility Criteria

Inclusion and exclusion criteria were established for the search.

The inclusion criteria were as follows:Articles published between 2015 and March 2022.Articles in English and Spanish.Cancer population and survivors. Articles whose sample consisted of persons with cancer at the time of the study or cancer survivors were examined.Individuals 18 years of age or older.Empirical studies.Studies that included measures of self-perceived or biomarker-derived physical health. Following Jim et al. [39], physical well-being, functional well-being, and self-assessed physical symptoms were utilized as measures of physical health. In addition, the presence/absence of biomarkers that provide an objective measure of the health status of individuals was added as a measure of physical health.

Based on these, the exclusion criteria were:7.Population related to cancer but who were not patients or survivors: caregivers, family members, nurses.8.Individuals under 18 years of age.9.Review studies, theoretical articles, and case studies.10.Studies whose only measure of R/S was religious and/or spiritual interventions. These types of studies were excluded as they did not apply to our objectives, since we were attempting to identify specific R/S variables that have an impact on physical health.11.Studies that included only mental health measures.

### 2.3. Selection Process

First, a review protocol was drawn up that allowed us to have a structured plan of the entire process, thus attempting to avoid biases that may appear during the process, as well as ensuring the transparency of the review. This step specified the need for the study, the review question, the search strategy, the eligibility criteria, the study quality assessment system, the data extraction method, data synthesis and dissemination.

After performing the search based on the above strategy, duplicate articles were eliminated through the ZOTERO bibliographic manager. The titles and summaries of all the records obtained were read. A checklist was created in Microsoft Excel, where it was determined whether each inclusion criterion and exclusion criterion was met, as well as the relevance of the articles for our purpose.

Subsequently, the full papers pre-selected in the abstract reading phase were read and examined. Using the checklist created earlier, the inclusion and exclusion criteria were reapplied, selecting the articles to be included in the review.

### 2.4. Evaluation of the Methological Quality of the Articles

The quality of the selected research was analyzed using the McMaster University Critical Review Form for Quantitative Studies [42], although the criteria of Appelbaum et al. [43] and Levitt et al. [44] were checked as a complement. This evaluation tool was selected due to its suitability for various research designs. This form contains a total of 19 criteria, grouped into eight sections: purpose of the study, literature, sampling, evaluation of results, intervention, analysis of results, and conclusions/clinical implications. In this evaluation, it was decided to omit the three items referring to the intervention, since one of the exclusion criteria of this review was that the R/S variable used was only an R/S intervention. For this reason, a total of 16 criteria were used to evaluate methodological quality.

To be as precise as possible in this assessment, we adapted the scores of the original 19-criteria version for the 16 criteria used in our evaluation, establishing a proportionality relationship between the values. Thus, in our quality assessment, the score was as follows:Excellent: 16.Very good: 14 to 15.Good: 12 to 13.Acceptable: 10 to 11.Poor: 9 or less.

### 2.5. Extracted Data

In view of the inclusion and exclusion criteria, the research question and the objective of the study, the baseline information extracted and recorded from each article was as follows:Title, authors, year of publication and journal.Country in which the study was conducted.Study sample: number of participants, type and stage of cancer, and whether patient or survivor.Objective, research design and instruments used.Key variables: measures of R/S and physical health.Main results or findings of each investigation.

## 3. Results

### 3.1. Selected Articles

The PsycInfo search yielded a total of 101 articles, while PubMed yielded 341 results. After eliminating 62 studies that appeared to be duplications, 380 articles remained for the first reading of titles and abstracts. A total of 206 articles that met exclusion criteria or were irrelevant were discarded.

The full text of the 174 articles was analyzed. After applying the inclusion and exclusion criteria, a total of 148 articles were obtained that were either ineligible or met exclusion criteria. Finally, a total of 26 articles were included in the review. The entire selection process may be viewed in Figure 1.

Although there was no inclusion criterion regarding the type of methodology used, the 26 articles included in the review employed a quantitative methodology. Of these, 22 used a cross-sectional design, while the remaining four were longitudinal studies.

The articles were published primarily in health care, public health, psychology, and oncology journals. There was not a substantial difference between the number of articles published in any given year. The year 2017 revealed the highest number of publications, a total of five. From 2018 to 2020, four articles were published per year, while in 2015 and 2021, three articles were published. In 2016, only two of the articles included in the review were published, while in 2022 (as of March) one article was published.

In regard to the countries in which the research studies were conducted, more than 50% were conducted in the United States (a total of 14). In addition, two of the studies were conducted in China. Of the remaining studies, one article was included in the review from each of the following countries: Saudi Arabia, Jordan, Iceland, Lebanon, India, Japan, Brazil, Chile, Turkey. Additionally, one article consisted of an international study involving several countries.

We also examined the various types of cancer that appeared in participants in the examined studies. Of these investigations, seven focused on a single type of cancer and one of the articles did not specify the type of cancer of the participants. The remaining articles did not focus their research on a particular type of cancer but included participants with disparate diagnoses. Not all of the articles consulted reported on all the types of cancer present in the sample; thus, the number of articles which each type of cancer mentioned appeared were analyzed.

Breast cancer appeared in the majority of articles (15), followed by genitourinary cancers (prostate, bladder, kidney, etc.), which appeared in 10 of the studies. Two other widely represented cancers were lung cancer and colorectal cancer, which appeared in nine and eight articles, respectively. Gynecological cancers (uterus, ovary, endometrium, fallopian tubes, etc.), hematological cancers (leukemia, lymphomas, myelomas, etc.) and gastrointestinal cancers (not including colorectal cancer) each appeared in seven investigations. The cancers appearing in a smaller number of studies were head and neck (four) and skin (two) cancers.

### 3.2. Quality Evaluation

Taking as a reference the criteria indicated by Law et al. [42], the methodological quality of all the articles evaluated ranged from good to very good, with the lowest score being 12 and the highest 15. Of the 26 articles selected, 21 were of very good quality (between 14 and 15), while the remaining 5 were of good quality (between 12 and 13). Table 1 displays the methodological quality of each article.

### 3.3. R/S Variables, Health Variables and Main Results

#### 3.3.1. R/S Variables

In two articles, the R/S variables used were simply religiosity and spirituality [45,46]. In those cases, these variables involved patients’ own self-perception of their R/S assessed through a single item. The remainder of the articles developed the R/S concept through various dimensions.

In 16 of the studies included in the review, the measure of religiosity/spirituality assessed was spiritual well-being. In 13 of these studies, spiritual well-being was understood as consisting of the three dimensions used by the FACIT-Sp12: meaning, peace, and faith [47,48,49,50,51,52,53,54,55,56,57,58,59]. In two other studies, spiritual well-being was understood as consisting of other dimensions, specifically those that comprise the EORTC QLQ-SWB32: existential, relationship with self, relationship with others, and relationship with someone/something greater [60,61]. The last article using spiritual well-being as a measure of R/S studied this variable (spiritual well-being) in regard to relationship with self and others, religious/spiritual beliefs and practices, and existential [62].

The next most-frequently used R/S variable in the research was religious/spiritual struggles, which is examined in three of the articles [63,64,65].

However, in other cases, R/S is assessed through variables such as spiritual experiences, religious beliefs and practices, congregational support, or religious coping [66,67,68,69]. The latter represents an R/S variable present in the previous three articles mentioned. Finally, one of the articles used spiritual pain as an R/S variable [70].

#### 3.3.2. Physical Health Variables

Regarding the physical health variables assessed in the research, we found that the variables that appeared most frequently pertain to physical symptoms, such as pain or fatigue, in a total of 16 articles [46,49,50,52,55,56,60,62,63,64,65,67,68,69,70]. Symptoms are followed by physical and functional well-being, which appeared as a health variable in a total of ten articles [47,48,50,51,53,55,56,57,58,59]). Physical quality of life appeared in two studies as a health variable [45,70]. In addition, physical functioning was examined as a physical health variable in five articles [52,60,61,62,67]. Finally, one article specifically used physical health as the main variable [54].

While all of the above physical health variables were subjective in nature, as they were self-reported measures, it is noteworthy that two of the articles used biomarkers such as cortisol [67] or telomere length [66], which are objective physical health measures.

#### 3.3.3. Main Research Results

The R/S variables mentioned above showed a varied association with the different health variables. Firstly, generic R/S, which does not specify any dimension, was associated with lower levels of pain in a cross-sectional study [46]. In another cross-sectional study, Al Ahwal et al. [66] found no significant associations between R/S and telomere length, a biomarker representing an objective measure of physical health status. In addition, R/S was found to be associated with increased physical well-being longitudinally [45].

As already mentioned, spiritual well-being is the R/S variable that appeared most frequently in the articles included in the review. Likewise, it was observed that it appeared in various ways, depending upon the dimensions that the authors established as components of spiritual well-being.

Spiritual well-being, understood as a construct composed of meaning, peace, and faith, was related in various ways to the physical and functional well-being of oncology patients, offering similar results in some cases and disparate results in others. In several cross-sectional studies, all three dimensions were positively and significantly associated with physical well-being and functional well-being [47,50,55]. In fact, some studies also found significant associations between total spiritual well-being and physical and functional well-being [55,59]. In contrast, there were studies that found positive associations of these dimensions only with functional well-being [48,51], and others that did so with physical well-being [57].

The results from the various studies varied depending on the dimension analyzed. Cheng et al. [53] observed that all three dimensions were positively associated with functional well-being, although meaning was not associated with physical well-being and faith was, albeit in a negative direction. Several studies found that while meaning and peace were associated with both types of well-being, faith was associated only with functional well-being [58,59]. The latter studies had a cross-sectional design. In contrast, through a longitudinal study, Leeson et al. [56] found that meaning/peace predicted greater physical and functional well-being.

One longitudinal study examined spiritual well-being but focused on the physical health status of individuals with cancer, measured through the physical component of the SF-36, as a self-perceived physical health variable [54]. The authors of this study found that meaning and peace were longitudinally positively and significantly associated with health status, while faith was not associated with this variable.

Spiritual well-being (composed of meaning, peace, and faith) was also related to physical symptoms in cancer patients, with similar heterogeneous results. Meaning and peace were found to be negatively correlated with various symptoms (fatigue, pain, dyspnea, loss of appetite, etc.) according to different cross-sectional investigations [49,52,55]. Only Brown et al. [50] found no relationships between meaning and peace and physical symptoms. In contrast, faith varied in its influence on symptoms. In some cases, higher faith was found to be associated with a higher level of physical symptoms [49], while in other cases, the association was negative [52]. This is similar to the longitudinal study by Leeson et al. [56], which found that meaning/peace predicted less fatigue and pain over time, whereas faith was associated only with greater fatigue.

The purported influence of spiritual well-being on the physical health of individuals with cancer was also analyzed in several cross-sectional studies that conceived of spiritual well-being as composed of relational (with self, others, or someone or something higher) or existential dimensions [60,61,62]. In this regard, two of the studies observed no relationship between global spiritual well-being with functional and physical symptoms [61,62], while Chen et al. [60] did observe significant relationships between these variables. The various dimensions correlated positively with physical functioning and negatively with physical symptoms, with the exception of the dimension of relationship with something or someone higher, which was not associated (or in the case of physical functioning, was negatively associated).

Another major variable, R/S struggles, was studied cross-sectionally in relation to physical symptoms. Two studies in our review found no significant associations between R/S struggles and pain [64,65]. In contrast, Damen et al. [63] did observe a positive association between symptomatic burden and R/S struggles. A single study, conducted by Pérez-Cruz et al. [70], evaluated spiritual pain, understood as “a deep pain in your soul/being that is not physical” (p. 2), observing that it was associated with lower physical quality of life and a higher number of symptoms.

Finally, a few studies focused on behavioral R/S variables. Among them, R/S coping stood out. On the one hand, several cross-sectional studies observed associations between negative coping and greater physical discomfort, considered as bodily discomfort related to physical weakness and disability [68], as well as worse physical functioning and more pain [67]. On the other hand, other behavioral aspects such as positive spiritual experiences and private religious practice were associated with better physical functioning and lower pain and, in addition, the former were also associated with lower cortisol levels [67]. In addition, a longitudinal study found that negative coping led to more sleep problems, as well as finding that private religious engagement was associated with more cancer-related symptoms [69].

Table A1, included in Appendix A, provides a detailed description of each study, linking the objectives, measures, and results with reference to the type of design and the main characteristics of the sample.

## 4. Discussion

This study aimed to detect the primary religious and spiritual variables considered in recent research in relation to the physical health of oncology patients, as well as to summarize the main findings of such research.

We have observed that recent research in this field particularly focuses on spiritual well-being. Spiritual well-being is often described as “a dynamic and affective dimension of religion and spirituality that impacts the way that people experience, understand and live their lives” [71]. In this sense, Leung and Pong [72] state that this well-being is an indicator of spiritual health, understood as a condition that guides individuals to identify their meaning and purpose in life, based on connections with others and the transcendent. Thus, spiritual well-being is beginning to be considered as another element of people’s holistic health, along with physical and mental health [72,73]. We believe this represents one reason why the use of spiritual well-being as a measure of R/S in health contexts has been increasing in recent years. In addition, it is worth noting that important previous studies have already highlighted the reasons why it is appropriate to use spiritual well-being as a measure of R/S in oncology settings. For example, Peterman et al. [74] argue that spiritual well-being allows for the assessment of spirituality across a broad spectrum of religious traditions and for people who consider themselves spiritual but not religious. Other authors, such as Puchalski [75], express this importance by referring again to spiritual well-being as a determinant of the integral health of these patients and, therefore, of their quality of life.

Spiritual well-being is generally conceived as a multidimensional construct which includes meaning, peace, and faith [76]. Thus understood, in the articles included in the review, this well-being has been related to various physical health variables in a number of ways. It is observed that, in general, greater spiritual well-being is associated with better physical health outcomes in oncology patients. In any case, considering these dimensions of spiritual well-being, it should be noted that the results obtained for the faith variable oblige us to be cautious when drawing conclusions. In addition to being heterogeneous, in many cases they may be contradictory, as associations are found in different directions or, on occasion, no relationship is found between faith and the physical health of cancer patients. It is possible that the general difference between the results of meaning/peace and faith pertains to their conceptualization in the FACIT-Sp, the main instrument for measuring these constructs. As discussed in one of the early and major studies on this topic, one of the strengths of the FACIT-Sp is that the Faith subscale can measure a dimension of spirituality that overlaps with or is reinforced by religion, whereas the Meaning/Peace subscale measures a dimension that is more independent [74]. Thus, it seems that the results linked to faith may be contradictory due to their association with religion. In this sense, although religion in many cases has positive effects on health, it may at times have a detrimental influence on it, as observed in one of the longitudinal articles included in the review [56]. Another possible reason is found in the research of Saiz et al. [77]: simple membership in a religious doctrine, regardless of spirituality, is not a reliable predictor of health-related benefits. In other words, faith without a daily practice and spiritual life may have counterproductive effects on health [77].

Spiritual well-being has also been defined as a construct composed of relational (with self, with others, with something or someone higher) or existential dimensions [60,61,62]. In general, spiritual well-being as measured cross-sectionally by these dimensions has been found to be positively associated with physical health of cancer patients, with certain exceptions concerning the dimension relationship with someone or something higher, which is not associated with physical health and, on one occasion, is negatively associated with physical functioning.

Thus, it may be important for research to attempt to unify the concept of spiritual well-being in order to understand how it influences or is associated with people’s physical health. As already mentioned, mental and physical health are closely interconnected, although they are generally conceived as different aspects; together with these, spiritual well-being appears as another element of health that must be considered. In this sense, considering the “mental” as a function of the physical organism may be helpful in explaining the relationship between spiritual well-being and physical health. In this regard, the research by Sleight et al. [57] included in this review sheds light on how spiritual well-being can attenuate the effect that some mental health problems have on physical health. Other included articles, such as the one by Kamijo and Miyamura [55], show how spiritual well-being is associated not only with physical and functional well-being, but also with mental and social well-being. Thus, following the previous line of reasoning, we could assume that spiritual well-being may be related to physical health through the influence of various aspects related to psychosocial well-being. Current research on religion, spirituality, and physical health therefore needs to focus on the study of psychosocial determinants (e.g., psychological well-being and social well-being) that influence these relationships, in order to understand the mechanisms by which R/S impacts physical health. However, while more research of this type is needed, there is the possibility that R/S acts on people’s health through as yet unknown pathways, which could include some form of divine intervention. This may be an option that researchers in any field should keep open. In any case, these avenues of research would represent a breakthrough in treating not only physical, but also holistic, health.

In addition, it seems necessary to emphasize those dimensions that imply a relationship with the transcendent. In the case of the faith dimension of the FACIT-Sp12 and the relationship with someone or something higher dimension of the EORTC QLQ-SWB32, it is observed that at times the results are either not significant or go in the opposite direction of better physical health. Both dimensions refer to the individual’s belief in God (i.e., it may have a religious connotation) or to spiritual beliefs. That is, they would belong to what Jim et al. [39] call the cognitive dimension of R/S. In this regard, the findings of Jim et al. [39] are congruent with the present review, as they do not observe a significant relationship between R/S beliefs and physical health in individuals with cancer.

Another variable that has been related to physical health in the review (specifically physical symptoms), specifically in a cross-sectional manner, is R/S struggles. In general, either no relationship was observed between the two variables, or a higher level of R/S struggles was found to be associated with a higher symptom burden. As with R/S struggles, there are other religious/spiritual variables that also involve a confrontation with the divine or with one’s own practices and beliefs, such as spiritual pain, which is also associated with poorer physical health in oncology patients [70]. In view of this possible relationship between R/S struggles or spiritual pain and physical health, the need to attend to and care for the spiritual needs of oncology patients becomes evident. In fact, several studies have highlighted the importance of spiritual care as part of the holistic health care of individuals [78,79].

In general, the important role of R/S in the psychosocial adjustment of individuals to cancer diagnosis and treatment has been highlighted, with a view to improving physical and mental well-being [80], through aspects such as religious coping. Nevertheless, we believe that what was observed in several of the articles in this review brings to light the need to address the negative R/S aspects present in oncology patients, such as R/S struggles or spiritual pain, within clinical and health care settings. This idea is related to the negative association that certain R/S variables have with physical health, mentioned above. Likewise, much research has been devoted to analyzing the relationships between variables of this type, such as R/S struggles [81,82] or mistrust in God [83] in relation to mental health. Therefore, we consider it necessary to further explore the effect on the physical health of oncology patients of R/S variables that imply a conflicting relationship with the divine or spiritual or a doubt about one’s own beliefs.

Be that as it may, these last-mentioned variables, together with others such as meaning or peace, would compose the affective dimension of R/S proposed by Jim et al. [39]. Thus, as with these authors, our review shows that this type of variable is generally associated with physical health.

Finally, several studies focus on a more behavioral dimension of R/S and its relationship with cancer, as Jim et al. [39] would term it, which likewise present heterogeneous results. Among these aspects we can highlight R/S coping, religious engagement, spiritual experiences, or religious practices [67,68,69]. In any case, these behavioral aspects of R/S are to a certain extent contradictory to the synthesis by Jim et al. [39], who did not observe an overall significant relationship between aspects such as coping or religious practice and physical health. However, for example, one of the longitudinal studies [69] has shown that certain behavioral aspects (such as negative coping or private religious engagement) have a negative impact on the physical health of oncology patients.

It is worth noting that, although R/S is generally studied in relation to perceived physical health, the cross-sectional study by Hulett et al. [67] represents one of the few attempts to test how R/S is associated with physical health objectively, through biomarkers that reflect the correct or incorrect functioning of the organism, such as cortisol itself. In the case of studies on the relationship between R/S and biomarkers, it also happens that a significant number of them are focused on biomarkers related to mental health, such as Posterior EEG Alpha (a putative biomarker of clinical outcome in major depression) [84]. Thus, and in view of the effects that R/S has on various aspects of physical health, more research is needed to understand the relationships between R/S and neuroendocrine and immune functions as represented by biomarkers to the extent that these, as proposed by Koenig et al. [1], affect susceptibility to, and recovery from, disease. Research of this type may be the way forward to definitively understand how the relationships between R/S and individuals’ health behave, although it requires a complex methodology that makes it difficult to carry out in many cases.

This need is evident in another of the studies in this review, which focuses on a particular aspect of R/S, attempting to associate it with telomere length, another biomarker representing an objective measure of physical health status [66]. Furthermore, this non-specific and general R/S has been associated with better physical health both longitudinally [45] and cross-sectionally [46]. Similarly, Jim et al. [39] observed that R/S measured in a general way (i.e., through single items that properly measure religiosity and/or spirituality), without focusing on affective, cognitive, or behavioral aspects, is associated with better physical health in oncology patients.

### Limitations and Future Implications

In any case, this review of the literature on the relationships between R/S and physical health in individuals with cancer has a number of limitations.

The main limitation involves the cross-sectional design used by the majority of studies on religion, spirituality, and physical health. This is reflected, therefore, in the fact that 22 of the 26 articles that have been included in the review present this type of design. As noted above, the problem with cross-sectional studies is that they do not allow us to infer causality, but only whether or not there is an association between variables. In this sense, cross-sectional data obtained in these types of studies may be misinterpreted as causal inferences (when they should be interpreted in terms of association), which leads to further increase in the heterogeneity of the results. For this reason, it is worth highlighting the need for new studies that address the relationships between religion, spirituality, and health through longitudinal or other RCT-like epidemiological designs. However, in the case of individuals with certain diseases, it is even more complicated to perform long-term studies. To mitigate the difficulty of this type of study in populations with a real risk of death, the time periods for follow-up of the measure could be limited and less invasive forms of data collection than those usually used could be established. In this regard, a protocol for a cohort study involving cancer patients in a longitudinal setting to analyze the relationship between R/S and health in this population has recently been published [85], which could be considered as a reference for a design that can reduce the shortcomings of cross-sectional designs. Nevertheless, despite the weaknesses of the cross-sectional designs discussed by Wang and Cheng [21] and mentioned in the introduction, all articles that met the defined inclusion criteria were included in the sample.

In addition, it is difficult to synthesize the results of studies with these characteristics. The variety of measures used is an obstacle to the use of other methods such as meta-analysis. In fact, the authors are generally transparent when defining the constructs that they deal with in their articles, thus making it possible to observe the disparity (or in some cases, the similarity) between one and the other. In many cases, the same variable is measured in different ways and through a variety of dimensions. A very clear case is that of spiritual well-being, which is at certain times understood as composed of meaning, peace, and faith, and at other times as consisting of the relationship with self, the relationship with others, or the relationship with someone or something higher and existential aspects. It occurs in a similar manner, for example, with R/S struggles. At times, these are understood as a form of negative coping, while at other times they involve a number of dimensions derived from the focus of conflict, such as struggles with the divine; struggles with the demonic; interpersonal struggles; moral struggles; struggles with doubts; or struggles with ultimate meaning. Therefore, we believe that a consensus is needed to conceptualize certain aspects of R/S closely related to health as proposed by Koenig [37], which may assist in improving holistic interventions that consider the religiosity and spirituality of oncology patients. In addition, this consensus would allow the distinction between affective, cognitive, and behavioral aspects of R/S proposed in meta-analytic reviews such as Jim et al. [39], which may assist in the replicability of this type of studies.

Finally, we would like to highlight sample and cultural diversity as a limitation that may affect the heterogeneity of results. Although most of the studies have been conducted in the United States, a large number of the studies have been carried out in very diverse cultural contexts and have used very different and on occasion too small sample sizes. More studies are needed that allow us to test for differences between cultures, as well as using similar large samples to facilitate generalization of the findings.

In any case, through this review we have been able to recognize the main religious and spiritual variables that are associated (in various ways) with different measures of physical health in cancer patients. We consider that the identification of these types of variables represents a step forward in the understanding of the relationships between R/S and the health of individuals with cancer. As we previously proposed, if we know which elements of R/S affect physical health and how these elements are related to other psychosocial aspects that present during the oncological process, we will have an increased likelihood of understanding the mechanisms by which R/S impacts the physical health of these patients.

## 5. Conclusions

This review summarizes the results of research on the relationships between R/S and physical health of cancer patients in recent years. It has been observed that the various R/S variables (spiritual well-being, R/S struggles, religious coping or self-rated R/S, among others) examined in the review generally displayed a positive association with the physical health of these patients. However, occasionally, R/S may be negatively associated with or unrelated to health. Our findings may assist in the treatment of various aspects of religiosity and spirituality, whether positive or negative, in health care contexts in order to improve the well-being and quality of life of individuals affected by cancer.

## Figures and Tables

**Figure 1 healthcare-10-01447-f001:**
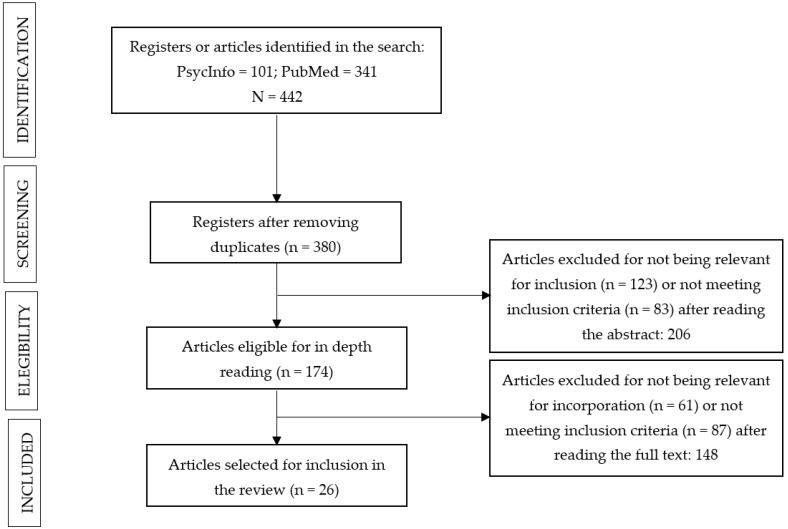
Article selection process.

**Table 1 healthcare-10-01447-t001:** Methodological quality of the articles included in the review.

Authors	Total (%)	Methodological Quality
Al Ahwal et al., 2018	14/16 (87.5%)	Very good
Al-Natour et al., 2017	15/16 (93.75%)	Very good
Asgeirsdottir et al., 2017	12/16 (75%)	Good
Bai et al., 2016	13/16 (81.25%)	Good
Best et al., 2015	15/16 (93.75%)	Very good
Brown et al., 2015	14/16 (87.5%)	Very good
Canada et al., 2016	15/16 (93.75%)	Very good
Cannon et al., 2022	15/16 (93.75%)	Very good
Chaar et al., 2018	14/16 (87.5%)	Very good
Chen et al., 2021	15/16 (93.75%)	Very good
Cheng et al., 2019	15/16 (93.75%)	Very good
Damen et al., 2021	14/16 (87.5%)	Very good
Gielen et al., 2017	12/16 (75%)	Good
Goyal et al., 2019	15/16 (93.75%)	Very good
Hulett et al., 2018	15/16 (93.75%)	Very good
Kamijo and Miyamura 2020	15/16 (93.75%)	Very good
King et al., 2017	15/16 (93.75%)	Very good
King et al., 2018	15/16 (93.75%)	Very good
Leeson et al., 2015	15/16 (93.75%)	Very good
Mendoça et al., 2020	15/16 (93.75%)	Very good
Narayanan et al., 2020	15/16 (93.75%)	Very good
Pérez-Cruz et al., 2019	12/16 (75%)	Good
Rohde et al., 2019	14/16 (87.5%)	Very good
Sleight et al., 2021	14/16 (87.5%)	Very good
Walker et al., 2017	13/16 (81.25%)	Good
Yilmaz and Cengiz 2020	14/16 (87.5%)	Very good

## Data Availability

The study did not report any data.

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
