# Peer review of "What Aspects of Religion and Spirituality Affect the Physical Health of Cancer Patients? A Systematic Review"

_healthcare, 2022, doi:10.3390/healthcare10081447_

Round 1

Reviewer 1 Report

Dear Authors,

Religion may be someone's personal choice, however, there is a close relationship between spirituality and positive mental health which can further contribute to the wellness of cancer patients. Although their mental health has been well discussed it will provide a complete glance at patients' overall wellness. 

As a suggestion, this psychological/mental health variable should be included in the review.

As we know that spirituality in a way provides a positive attitude toward life and the positive attitude can be a predictor of positive health outcomes. 

Hence, including psychological or mental health studies would make this study more generalized and powerful.

Suggestion

1. Statement mentioned in lines number 98-101 (It is always useful to review the most recent work in a field of study. Additionally, during the COVID-19 pandemic, new approaches and studies have emerged that are expanding research on R/S, and health research, insofar as this virus has been a major stressor), is not required.

You may either remove it or have a genuine explanation. 

Author Response

Response to Reviewer 1 Comments

Dear Authors,

Religion may be someone's personal choice, however, there is a close relationship between spirituality and positive mental health which can further contribute to the wellness of cancer patients. Although their mental health has been well discussed it will provide a complete glance at patients' overall wellness. 

As a suggestion, this psychological/mental health variable should be included in the review.

As we know that spirituality in a way provides a positive attitude toward life and the positive attitude can be a predictor of positive health outcomes. 

Hence, including psychological or mental health studies would make this study more generalized and powerful.

Dear reviewer, thank you very much for your review and recommendations. We appreciate your willingness to contribute to the improvement of our manuscript. Regarding your comments, we will respond to you below:

We fully agree that adding mental health variables would make the study more powerful. We have taken your proposal into consideration and, in fact, we have added a paragraph in the introduction with which we try to indicate the relationship between mental and physical health (lines 96-100). In it, we expose the difficulty in separating the two types of health, insofar as both interact and, therefore, mental health has a great impact on physical (and general) health. Likewise, the discussion has added a debate on the possible relationships between physical and mental health insofar as these may be influenced by spiritual well-being (lines 465-481).

However, our decision to focus on physical health indicators is due to several reasons. One of them lies in the fact that there is a majority of studies that focus on mental health, as exposed by Koenig (2012). We believe that this overwhelming difference in the number of items focused on mental health could affect the outcome on physical health indicators.

Likewise, there are several reviews that focus on the relationships between religion, spirituality and mental health in cancer patients (DOI: 10.1002/cncr.29350; DOI: 10.1016/j.ctim.2016.02.012). We believe that the effort should be directed along these lines, that is, to focus on continuing to explore these relationships through reviews and meta-analyses, in the same way that we have tried to do with physical health. The large volume of research would greatly complicate the task.

Perhaps, in future studies, it might be interesting to conduct a systematic review of studies dealing with physical and mental health, to observe how the two interact with each other.

In any case, we greatly appreciate your comment. It has undoubtedly raised a debate that we consider important to address and that in the future will lead us to implement its proposals. As you propose, mental health research needs such a study.

Suggestion

  1. Statement mentioned in lines number 98-101 (It is always useful to review the most recent work in a field of study. Additionally, during the COVID-19 pandemic, new approaches and studies have emerged that are expanding research on R/S, and health research, insofar as this virus has been a major stressor), is not required.

You may either remove it or have a genuine explanation. 

We believe that you are absolutely right. We believe that approaching the COVID problem is not appropriate, since the article does not deal with it and the discussion does not refer to it. We have therefore decided to accept your proposal to delete this part of the text. We are very grateful for this proposal, which undoubtedly improves the quality of the manuscript.

Reviewer 2 Report

Brief summary:

First, I thank the editor for the opportunity to review this systematic review of the associations between R/S and physical health measures in cancer patients.

The authors have done a great job in preparing the paper, and I commend them for their efforts.

I really enjoyed reading the paper, the introduction was superb with great clarity and depth related to the definitions pertaining to the field. It was easy to follow, and the composition and language were pleasant. The paper deals with an important topic and the syntheses of results made in the field of spirituality and health, such as this paper aspires, are much needed to make this important knowledge accessible to health policymakers and clinicians.

I have below included some general comments and some specific comments to guide the authors in revising the paper, and hope that they will be of some help.

General concept comments

I find the major issue to be the lack of reflection and discussion of the level of evidence that is provided by the utilized approach. The authors primarily include articles with cross-sectional data in the review which (in by far most cases) are generally not able to provide evidence for causal inferences – which is basically the type of inferences we need to make at this point in the field of spirituality, religion, and health. The issue of low-quality evidence in many studies due to small samples and cross-sectional designs should be addressed in both the background and in the discussion.

Longitudinal studies are important to clearly distinguish causal structures. At the most, the review will be able to hint at (for objective 2) which R/S variables are “associated” with physical health. I believe the current and contemporary research in the field should address this by heightening the methodological standards from that of cross-sectional to longitudinal or other RCT-like epidemiological designs. It could be proposed to either exclude cross-sectional studies or at least stratify your results based on this difference in method (cross-sectional or longitudinal) as the method quality evaluation (after Law et al.) doesn’t seem to rightfully reflect this difference. It is unclear to the reader which results are from what kind of studies. The reason for including cross-sectional articles should be outlined.

The authors argue that prior research mostly documents the positive mental effects of R/S, using this as an argument for the rationale of their scope, but rather fail in providing compelling evidence that this is so. Much methodologically sound research has been conducted on physical health effects as well. More extensive referencing to document the past research on the effect of R/S on both mental and physical health (and even social) should be provided to clarify that a review of the effects of R/S on physical health is warranted. See e.g. :

Koenig, H. G., King, D. E., and Carson, V. B. (2012), Handbook of religion and health (2nd ed.), Oxford / New York: Oxford University Press.

Li, S., Stampfer, M. J., Williams, D. R., and VanderWeele, T. J. (2016), "Association of Religious Service Attendance With Mortality Among Women," JAMA Intern Med, 176 (6), 777-785. DOI: 10.1001/jamainternmed.2016.1615.

Chen, Y., Kim, E. S., and VanderWeele, T. J. (2021), "Religious-service attendance and subsequent health and well-being throughout adulthood: evidence from three prospective cohorts," Int J Epidemiol, 49 (6), 2030-2040. DOI: 10.1093/ije/dyaa120.

The authors, having prepared a review protocol could have preregistered it e.g. at PROSPERO or other similar platforms. Without this step, there is no way to assess the quality of this protocol. Although not relevant at this point, it is noted for similar work in the future.

In the discussion, it would be very relevant to discuss the potential interpretations of the heterogeneity of the findings further with respect to the above-mentioned issues with the methodological limitations of cross-sectional studies on this topic. The authors are finding quite heterogeneous results across the included studies – is this due to the fact that the effects of R/S on health vary a lot? Or could it be due to faulty causal inferences drawn in the papers included based on cross-sectional data where longitudinal data had been needed? Or could it be due to cultural or sample differences? Or specific disease prognoses and trajectories?

The paper deals with R/S issues/struggles/pain as well and doesn’t mention spiritual care as the act of caring for such spiritual needs or struggles. Mentioning spiritual care and needs in the discussion would perhaps improve the clinical relevance of the paper. See e.g.:

Hvidt, N. C., Nielsen, K. T., Korup, A. K., Prinds, C., Hansen, D. G., Viftrup, D. T., Assing Hvidt, E., Hammer, E. R., Falko, E., Locher, F., Boelsbjerg, H. B., Wallin, J. A., Thomsen, K. F., Schroder, K., Moestrup, L., Nissen, R. D., Stewart-Ferrer, S., Stripp, T. K., Steenfeldt, V. O., Sondergaard, J., and Waehrens, E. E. (2020), "What is spiritual care? Professional perspectives on the concept of spiritual care identified through group concept mapping," BMJ Open, 10 (12), e042142. DOI: 10.1136/bmjopen-2020-042142.

Büssing, A. (2021), "The Spiritual Needs Questionnaire in Research and Clinical Application: a Summary of Findings," J Relig Health, 60 (5), 3732-3748. DOI: 10.1007/s10943-021-01421-4.

In appendix A, please include countries and sample characteristics (maybe even age/gender compositions) so that the study settings are more easily discernable.

Specific comments

P 2 L 82: should add “[…] spiritual impact” – after all, you are considering R/S in relation to cancer.

P 3 L 110-112: “the associated psychosocial variables” - associated with what? Aspects of R/S or physical health? slightly unclear, please rephrase.

P3 L 125: affect the “physical health” of cancer patients? This aim would be clearer is rephrased, as the authors state that they are interested in physical health rather than mental health as the latter is more strongly examined.

P3 L 129: This seems like too steep a promise, as you can hardly avoid introducing bias altogether by any means. I would suggest rephrasing to “… thus minimizing the potential bias introduced in the conclusions” or something similar.

P3 L 142-44: This is too unspecific. What is e.g. meant by “the corresponding Boolean operators”. The procedure rather makes "reproduction" possible than “objectifies” it. Further, it is mentioned that the terms were used as “reference” – this seems unclear to me. Were they used as search terms? Please provide the specific search strings used.

P4 L 182: Figure 1 is referenced in the result section and may be omitted from here.

P6 L 257: “were conducted in the US”. rather than was.

P6 L 272: “were lung cancer…”

P7 L 288: “had to do”

P11 L 514: with so many of the included studies being cross-sectional it is simply not (in most cases) possible to conclude the direction of causal effects. As such, it doesn’t seem reasonable to say that the authors have been able to “recognize … variables that do or don’t influence … physical health”. I would suggest specifying that you have identified variables that are associated (in various ways) with physical health measures.

P11 L 513-16: These two sentences seem redundant. Please amend.

P12 L 528-29: Again, I would suggest speaking of associations rather than “impact” which implies causal inferences that you probably cannot draw from the material.

Author Response

Response to Reviewer 2 Comments

Brief summary:

First, I thank the editor for the opportunity to review this systematic review of the associations between R/S and physical health measures in cancer patients.

The authors have done a great job in preparing the paper, and I commend them for their efforts.

I really enjoyed reading the paper, the introduction was superb with great clarity and depth related to the definitions pertaining to the field. It was easy to follow, and the composition and language were pleasant. The paper deals with an important topic and the syntheses of results made in the field of spirituality and health, such as this paper aspires, are much needed to make this important knowledge accessible to health policymakers and clinicians.

I have below included some general comments and some specific comments to guide the authors in revising the paper, and hope that they will be of some help.

Dear reviewer, thank you very much for your review, your recommendations and your kind words. We appreciate your willingness to contribute to the improvement of our manuscript. Regarding your comments, we will respond to you below:

General concept comments

I find the major issue to be the lack of reflection and discussion of the level of evidence that is provided by the utilized approach. The authors primarily include articles with cross-sectional data in the review which (in by far most cases) are generally not able to provide evidence for causal inferences – which is basically the type of inferences we need to make at this point in the field of spirituality, religion, and health. The issue of low-quality evidence in many studies due to small samples and cross-sectional designs should be addressed in both the background and in the discussion.

Longitudinal studies are important to clearly distinguish causal structures. At the most, the review will be able to hint at (for objective 2) which R/S variables are “associated” with physical health. I believe the current and contemporary research in the field should address this by heightening the methodological standards from that of cross-sectional to longitudinal or other RCT-like epidemiological designs. It could be proposed to either exclude cross-sectional studies or at least stratify your results based on this difference in method (cross-sectional or longitudinal) as the method quality evaluation (after Law et al.) doesn’t seem to rightfully reflect this difference. It is unclear to the reader which results are from what kind of studies. The reason for including cross-sectional articles should be outlined.

We fully agree with this comment. We have given the manuscript another revision considering this. Indeed, the use of causal expressions in non-experimental studies can induce readers into assuming a cause–effect relationship when alternative explanations have not been ruled out. Even when primary studies fall into this questionable practice, we do not wish to perpetuate it. A proper language and few disclaimers along the text should suffice to prevent misleading readers. For instance, in the abstract we changed "...R/S seems to have an influence on the physical health..." to "...R/S seems to have a positive association with the physical health...".

However, the selection of cross-sectional studies is not by choice. The aim of our systematic review was to find any study that met all the inclusion criteria (and none of the exclusion criteria). Methodology was neither an inclusion nor an exclusion criterion.

As the reviewer states, it would be convenient to make casual inferences in the field of R/S and health. The inferential superiority of experiments or quasi-experiments over non-experimental designs has been well argued. Logitudinal studies may give some clues about the causal relationships between variables, but they are not enough to do so with guarantees in absence of experimental controls and random assignment. In natural or clinical settings, of course, it may be impossible to design an experimental (or even quasi-experimental) study. Sometimes, manipulation of the independent variable is not possible (or ethical) and a non-experimental methodology is the only means to study the relationship among variables. The clinical setting makes difficult even to extract data in a longitudinal design, even when it would be a non-experimental one. We can argue this is the reason why the vast majority of primary studies are non-experimental and cross-sectional (and the rest are longitudinal but also non-experimental). However, multiple non-experimental studies (cross-sectional or otherwise) can contribute (they usually do) to accrue evidence supporting causal hypothesis. That is one of the advantages  of a systematic review including many studies on a specific topic, even when they depict non-experimental designs.

For the sake of clarity, we have ordered the studies according to their design, so that within each R/S measure discussed in the results we have differentiated between cross-sectional and longitudinal studies. Similarly, we have applied this differentiation in the discussion, although we have noted that in some cases the use of the term influence or impact is well done (for example, in the part of the discussion where it is said that sometimes a detrimental influence of religion on health has been observed). In these cases, we have also added the type of design used to further clarify the scope of the findings. We have mentioned this problem in the introduction (as well as the limitations of small samples), through a subsection where we state that there is a large part of the literature that uses cross-sectional designs (with some examples) and explaining the main limitations of this type of design. In addition, we no longer speak of "influence" but of "association" in all appropriate cases.

The issue of cross-sectional designs has been added as a limitation, and the need for longitudinal studies in this field has also been proposed. Here we have also added a reason why it was necessary to include cross-sectional studies in our sample: despite knowing the weaknesses of these designs, we decided to include all articles that met the defined inclusion criteria. In addition, we have alluded to the difficulty of elaborating longitudinal studies in samples of cancer patients, as well as a brief reflection on how this type of design could be facilitated.

We believe that through theese changes the reader will be able to distinguish which R/S variables are associated with the physical health of oncology patients, and in which cases the former have an influence on the latter. We hope that this will improve the quality of the manuscript. Thank you again for this important observation.

The authors argue that prior research mostly documents the positive mental effects of R/S, using this as an argument for the rationale of their scope, but rather fail in providing compelling evidence that this is so. Much methodologically sound research has been conducted on physical health effects as well. More extensive referencing to document the past research on the effect of R/S on both mental and physical health (and even social) should be provided to clarify that a review of the effects of R/S on physical health is warranted. See e.g. :

Koenig, H. G., King, D. E., and Carson, V. B. (2012), Handbook of religion and health (2nd ed.), Oxford / New York: Oxford University Press.

Li, S., Stampfer, M. J., Williams, D. R., and VanderWeele, T. J. (2016), "Association of Religious Service Attendance With Mortality Among Women," JAMA Intern Med, 176 (6), 777-785. DOI: 10.1001/jamainternmed.2016.1615.

Chen, Y., Kim, E. S., and VanderWeele, T. J. (2021), "Religious-service attendance and subsequent health and well-being throughout adulthood: evidence from three prospective cohorts," Int J Epidemiol, 49 (6), 2030-2040. DOI: 10.1093/ije/dyaa120.

We believe that you are absolutely right in this respect. In trying not to overextend ourselves, we have not sufficiently justified our study on the basis of previous research. We have considered the suggested references and added them. In addition, we have expanded the range of references to both mental health and physical health research to provide the reader with even more empirical evidence of the need for such a study. Finally, we wanted to mention, as you suggested, social health in order to broaden the understanding of the influence of R/S on people's holistic health.

The authors, having prepared a review protocol could have preregistered it e.g. at PROSPERO or other similar platforms. Without this step, there is no way to assess the quality of this protocol. Although not relevant at this point, it is noted for similar work in the future.

We agree with you and regret not having preregistered our review protocol. Undoubtedly, such a step would have facilitated the evaluation of its quality. We have considered registering it now but, having examined PROSPERO's requirements for registration, we considered that it would not be appropriate. Under the circumstances, we would not meet the following PROSPERO requirements:

  • Registration should take place once the systematic review protocol has been finalised, but ideally before screening studies for inclusion begins. However, reviews are currently accepted for registration as long as they have not started data extraction.
  • Completed reviews should not be registered.

Again, we regret that we did not take this step, although we will take it into consideration for future revisions.

In the discussion, it would be very relevant to discuss the potential interpretations of the heterogeneity of the findings further with respect to the above-mentioned issues with the methodological limitations of cross-sectional studies on this topic. The authors are finding quite heterogeneous results across the included studies – is this due to the fact that the effects of R/S on health vary a lot? Or could it be due to faulty causal inferences drawn in the papers included based on cross-sectional data where longitudinal data had been needed? Or could it be due to cultural or sample differences? Or specific disease prognoses and trajectories?

Again, we fully agree with this comment. Undoubtedly, in line with previous comments, issues related to study design may have to do with the heterogeneity of results that we have mentioned as a limitation. We greatly appreciate this input, which will improve the quality of our discussion.

Given the above comments, we have added a paragraph mentioning the limitation of the scope of the results obtained from cross-sectional studies. In this paragraph we wanted to point out that cross-sectional studies only allow us to infer an association, not causality. In this regard, we have pointed out that the studies sometimes interpret the cross-sectional data erroneously, which may explain the heterogeneity of the results. Likewise, as we have already mentioned, we have highlighted the need for studies using longitudinal designs to avoid the limitations of cross-sectional designs. In any case, we have also mentioned the difficulties that may exist when doing longitudinal studies with samples of patients with life-threatening diseases, as well as a suggestion of how one could work in this regard.

In addition, we wanted to mention aspects such as cultural and sampling diversity, as you suggested. We believe that the scope of the study is now more realistic and remains relevant to the scientific community. Of course, we outline the need for more longitudinal approaches to research in this field. Thanks again for the interesting observation.

The paper deals with R/S issues/struggles/pain as well and doesn’t mention spiritual care as the act of caring for such spiritual needs or struggles. Mentioning spiritual care and needs in the discussion would perhaps improve the clinical relevance of the paper. See e.g.:

Hvidt, N. C., Nielsen, K. T., Korup, A. K., Prinds, C., Hansen, D. G., Viftrup, D. T., Assing Hvidt, E., Hammer, E. R., Falko, E., Locher, F., Boelsbjerg, H. B., Wallin, J. A., Thomsen, K. F., Schroder, K., Moestrup, L., Nissen, R. D., Stewart-Ferrer, S., Stripp, T. K., Steenfeldt, V. O., Sondergaard, J., and Waehrens, E. E. (2020), "What is spiritual care? Professional perspectives on the concept of spiritual care identified through group concept mapping," BMJ Open, 10 (12), e042142. DOI: 10.1136/bmjopen-2020-042142.

Büssing, A. (2021), "The Spiritual Needs Questionnaire in Research and Clinical Application: a Summary of Findings," J Relig Health, 60 (5), 3732-3748. DOI: 10.1007/s10943-021-01421-4.

Thank you very much for this comment. We consider it very accurate and, therefore, we have taken it into account and we have alluded in the discussion to the attention and care of spiritual needs, based on the references you propose (lines 497-501). This undoubtedly gives more clinical relevance to our work.

In appendix A, please include countries and sample characteristics (maybe even age/gender compositions) so that the study settings are more easily discernable.

We have assessed this comment and also consider it appropriate. For reasons of space, we have included the aspects you suggest in the sample column. We believe that this will make it easier for the reader to understand the context and population characteristics of each study. Thank you again for your constructive comments.

 Specific comments

P 2 L 82: should add “[…] spiritual impact” – after all, you are considering R/S in relation to cancer. Done, thank you.

P 3 L 110-112: “the associated psychosocial variables” - associated with what? Aspects of R/S or physical health? slightly unclear, please rephrase. Done, thank you.

P3 L 125: affect the “physical health” of cancer patients? This aim would be clearer is rephrased, as the authors state that they are interested in physical health rather than mental health as the latter is more strongly examined. Done, thank you.

P3 L 129: This seems like too steep a promise, as you can hardly avoid introducing bias altogether by any means. I would suggest rephrasing to “… thus minimizing the potential bias introduced in the conclusions” or something similar. Done, thank you.

P3 L 142-44: This is too unspecific. What is e.g. meant by “the corresponding Boolean operators”. The procedure rather makes "reproduction" possible than “objectifies” it. Further, it is mentioned that the terms were used as “reference” – this seems unclear to me. Were they used as search terms? Please provide the specific search strings used. Done, thank you.

P4 L 182: Figure 1 is referenced in the result section and may be omitted from here. Done, thank you.

P6 L 257: “were conducted in the US”. rather than was. Done, thank you.

P6 L 272: “were lung cancer…” Done, thank you.

P7 L 288: “had to do” Done, thank you.

P11 L 514: with so many of the included studies being cross-sectional it is simply not (in most cases) possible to conclude the direction of causal effects. As such, it doesn’t seem reasonable to say that the authors have been able to “recognize … variables that do or don’t influence … physical health”. I would suggest specifying that you have identified variables that are associated (in various ways) with physical health measures. Done, thank you.

P11 L 513-16: These two sentences seem redundant. Please amend.

We have deleted the second sentence and modified the first sentence based on the previous comment. By mistake we had repeated it but in a different way.

P12 L 528-29: Again, I would suggest speaking of associations rather than “impact” which implies causal inferences that you probably cannot draw from the material.

Thank you for these specific comments. We have corrected each one of them. They have certainly helped us to improve the grammatical quality of the writing and its comprehension.

Once again, we thank you for your willingness to constructively evaluate our work and thus help to improve its quality.

Reviewer 3 Report

Thank you very much. Please see my comments in the attached file.

Author Response

Response to Reviewer 3 Comments

Dear reviewer, thank you very much for your review, your recommendations and your kind words. We appreciate your willingness to contribute to the improvement of our manuscript. Regarding your comments, we will respond to you below: 

Review comments:

Section 1: In Lines 35-36 the assumption that a religious belief apparently inevitably leads to devotion and surrender is a conceptual bias, I think, as there are many traditions which are quite comfortable in a constant questioning and even resist (or allow for adherents who do resist) particularized devotion and/or surrender; Judaism in its non-Orthodox forms would be oneprominent example.

We believe this is a very sensible and accurate comment. We greatly appreciate that you have done so. It is absolutely true that this statement implies that all religious traditions uncritically accept dogmas and practices, and that only spirituality would allow a free approach. For this reason, we have considered that it is best not to make this assumption, so as not to establish false ideas and not to confuse the reader with intricate explanations.

Line 42: Do you mean “immanent absolute”?

Yes, it was a typo. It has been corrected. Thank you very much for pointing it out.

Lines 50-51: By the clinical view, is this a reference to psychological perspectives whereby the human psyche, in biological function, has naturally occurring “spiritual” elements (that is, taken as such by the experiencer and not empirically speaking)? Some emendation may help the reader to follow this.

This is another very accurate comment. We believe that this phrase failed to express our idea correctly. We want to refer to something like attention to the spiritual dimension from a clinical point of view. In other words, spirituality understood as an element that healthcare professionals can consider in their approach to improving the well-being of patients. We have changed this sentence so as not to enter into perspectives that may be confusing to some readers. For a better understanding, moreover, we have provided an example from the mental health context through a reference (Moreira-Almeida et al., 2016). We hope this will solve any possible confusion that may arise. We greatly appreciate the comment.

Section 2: Introduction: A brief summary of the PRISMA guide could be useful for we readers unfamiliar with its contents.

We fully agree. We have added several sentences briefly describing what the PRISMA guide is and its usefulness, using language that we believe is appropriate for readers unfamiliar with the guide to understand what it is all about.

Line 106ff: As these appear to be subcriteria further indentation or numerals instead of the same aligned bullet points might make it easier to follow.

We agree with this comment. However, as we are following the format of the journal, we cannot enlarge the indentation of the text. On the other hand, the format does allow us to use numbered lists, so we have changed the bullets points for numbers.

Section 3: Lines 219, 258, 269, 271, 275, and 289: “were analyzed”, “were conducted”, “were analyzed”, “which appeared”, “each appeared”, “articles developed”, respectively (apologies on being overly detailed…)

Thank you for pointing out these grammatical errors. They have all been corrected.

3.3.2 and 3.3.3: These paragraphs also need to be put into the past tense for reporting on what the findings were.

Thank you again for pointing out these errors. As in the previous case, they have been corrected so that all the text is in the correct verb tense.

Can “spiritual well-being” be considered an aspect of physical health? Some justification or explanation of how it might affect physical factors might be good to include here, even parenthetically. The connections mentioned between this, physical well-being, and functional well-being were interesting. (The paragraph of Lines 350-357 is very good in this regard; perhaps other parts of this section might be worded more like it? I assume the Discussion section will cover such and so this is merely a suggestion.)

Section 4: I agree that spiritual well-being seems a clear candidate for holistic health measures, but as the title of thearticle specifically relates to physical healthI think it might be helpful to draw clearer analytical lines for how “mental”is differentiated from “physical”if we take the mental to itself be a function of the physical organism. Doing this could make it clearer how a rather aethereal notion such as “spiritual well-being”may impact upon factors like dealing with pain and discomfort.

Lines 437-438: Yes, precisely this!

We fully agree with these last comments and thank you very much. In line with what was proposed by another reviewer, we wanted to add a paragraph in this regard in the introduction (lines 96-100). In it we mentioned the difficulty that often exists in separating the physical from the mental, insofar as both aspects are clearly interconnected. We believe that this gives rise to a better understanding of the discussion.

Also, in the discussion we wanted to reintroduce this idea, in order to give a possible explanation of how spiritual well-being can influence physical well-being through mental well-being. To this end, we have shown how the different types of well-being that are usually considered are associated with each other.  In addition, we have taken into account the idea of "the mental as a function of the physical organism". We have added this idea and, in addition, we wanted to mention social well-being as a health factor that can also mediate such relationships. The subject of social well-being has also been mentioned briefly in the introduction, so that it is not left loose here. You can observe it in lines 465-481.

All this has led us to emphasize the need to understand the mechanisms underlying the relationships between R/S and health.

We believe that with the new elements that we have added, our conclusions are aimed at physical health, but without losing sight of the fact that this is one more component of the overall health of individuals. Again, thank you very much for the very accurate comments.

The paragraph of Lines 478-485 is very intriguing, and this is maybe one avenue that could be further explored or expanded even in the present review.

Thank you very much for your observation. We agree with you. We have done a detailed search for studies that have linked R/S to biomarkers, noting that in addition to being few, many are focused on mental health. We have added to the aforementioned lines some references to show how the use of biomarkers in studies on mental health is broader, while pointing out the difficulty of conducting studies of this type given the complexity of the methodology required. We believe that this broadens the idea a bit and opens the way to the need for further studies in this regard.

Lines 490-491: If “R/S measured in a general way”does not include items like “affective, cognitive or behavioral aspects”then what does it include?

Thanks for the observation. It is true that it is not entirely clear what "generic R/S" means. In some cases, studies work with single items that refer to questions such as "How religious do you consider yourself?". That is, they are self-report measures that do not specify whether you measure your religiosity based on your practices (e.g., how much you go to church), your attachment to God, your spiritual well-being, etc. We have added a sentence that we believe may help to clarify this.

The points in the Limitations section are good; indeed the definitional parameters seem badly in need of being made uniform. Perhaps some indication of how common it was for the articles studied to transparently outline how these terms were used by the authors within their own articles would help the reader get a better picture of how disparate or similar the field is at present.

Thank you for the comment. We have added a sentence to that effect in the discussion about the variety of measures and concepts used in the research.

Again, thank you very much for your willingness to constructively evaluate our work and thus help to improve its quality.

Round 2

Reviewer 2 Report

Brief summary

Dear editors and authors,

I thank the authors for their detailed, thoughtful, and relevant answers to all my comments. This was a very satisfying reading as it was clear that the authors had reflected constructively on the comments given. I am happy that my comments were helpful and commend them on their efforts in revising and improving the paper.

I believe that the paper has improved tremendously. It is now much clearer with the outwritten clarification of design in findings and the discussion is much more nuanced - lovely!

I have only a few comments left.

General concept comments

In my opinion, the comment introduced in the paper regarding that the “separation” of physical and mental health is arbitrary (L109), is ill put. I believe that such a claim would take immense ontological clarification and presentation, that is really not the aim of the paper. Generally, there are various ontological and epistemological differences based on scientific discipline etc. Even though these two domains interact, it is hardly an argument that they are the same. Different constructs may be correlated or affect each other both ways without being the same thing. In my humble opinion, I think it is paramount to keep these constructs separate. I would suggest that it could suffice to say something along the lines of: “since physical and mental health constructs have been shown to interact in numerous ways, it may be difficult to separate and identify the effective driving factors. Specific study designs, such as experimental RCTs or longitudinal epidemiologic design may help in distinguishing the direction of effects”.

You might want to cite the following recently published and very extensive review:

Balboni, T. A., VanderWeele, T. J., Doan-Soares, S. D., Long, K. N. G., Ferrell, B. R., Fitchett, G., Koenig, H. G., Bain, P. A., Puchalski, C., Steinhauser, K. E., Sulmasy, D. P., and Koh, H. K. (2022), "Spirituality in Serious Illness and Health," JAMA, 328 (2). DOI: 10.1001/jama.2022.11086.

The authors mention in the discussion that more research on psychosocial aspects of R/S should be made to determine the pathways through which R/S acts on health. The sentence sounds like the authors think that enough research on psychosocial factors in R/S would be able to explain all R/S effects on health(around L883). Albeit it is definitely true that more such research is needed, the authors may consider including a line or two on the possibility that R/S works on health through yet unknown paths that could include some form of divine interventions. While this is surely a bold statement, I believe that it from a scientific theory perspective is a possibility that researchers in all fields should keep open.

Specific comments

L173: the aim now has a spelling error… delete “may”

L240: I believe “search string” is more commonly used than “equation”, but I might be wrong?

L978: a recently published protocol of a cohort study includes cancer patients in a longitudinal setup to study R/S and health. You may consider referencing it as an example of a study and design that might mitigate the shortcomings of cross-sectional studies on R/S and health in cancer patients.

Stripp, T. K., Wehberg, S., Büssing, A., Andersen-Ranberg, K., Jensen, L. H., Henriksen, F. L., Laursen, C. B., Søndergaard, J., and Hvidt, N. C. (2022), "Protocol for EXICODE: the EXIstential health COhort DEnmark—a register and survey study of adult Danes," BMJ Open, 12 (6), e058257. DOI: 10.1136/bmjopen-2021-058257.

Appendix A:

For brevity, you may note only e.g. “52% female” and omit the percentage for males. The reader will be able to figure out that the rest (48%) are males – for all practical reasons anyway… The number of transsex/gender persons is usually very low in studies like these.

Island should be Iceland

Author Response

Response to Reviewer 2 Comments (Round 2)

Dear editors and authors,

I thank the authors for their detailed, thoughtful, and relevant answers to all my comments. This was a very satisfying reading as it was clear that the authors had reflected constructively on the comments given. I am happy that my comments were helpful and commend them on their efforts in revising and improving the paper.

I believe that the paper has improved tremendously. It is now much clearer with the outwritten clarification of design in findings and the discussion is much more nuanced - lovely!

I have only a few comments left.

Dear reviewer,

We greatly appreciate your new comments. We believe that each of them again contributes to the improvement of the quality of the manuscript. We would therefore like to thank you for your willingness to constructively evaluate our work, as well as for your kind words. We hope you like the final result.

General concept comments

In my opinion, the comment introduced in the paper regarding that the “separation” of physical and mental health is arbitrary (L109), is ill put. I believe that such a claim would take immense ontological clarification and presentation, that is really not the aim of the paper. Generally, there are various ontological and epistemological differences based on scientific discipline etc. Even though these two domains interact, it is hardly an argument that they are the same. Different constructs may be correlated or affect each other both ways without being the same thing. In my humble opinion, I think it is paramount to keep these constructs separate. I would suggest that it could suffice to say something along the lines of: “since physical and mental health constructs have been shown to interact in numerous ways, it may be difficult to separate and identify the effective driving factors. Specific study designs, such as experimental RCTs or longitudinal epidemiologic design may help in distinguishing the direction of effects”.

We agree with you in this opinion. It is true that calling this separation "arbitrary" opens up a range of possibilities that could give rise to a great deal of doubt in the reader as to "why this separation is arbitrary". Undoubtedly, it would require extensive clarification that would exceed the limits set by the purpose of this review. Therefore, we have considered that the phrases you propose are perfect for not generating this problem. Likewise, we wanted to leave the quote that followed the previous sentences, to give an example of the difficulty involved in trying to separate the physical from the mental. You can see this modification on lines 95-101. Thank you very much for your suggestion, we hope that you will consider the modifications appropriate.

You might want to cite the following recently published and very extensive review:

Balboni, T. A., VanderWeele, T. J., Doan-Soares, S. D., Long, K. N. G., Ferrell, B. R., Fitchett, G., Koenig, H. G., Bain, P. A., Puchalski, C., Steinhauser, K. E., Sulmasy, D. P., and Koh, H. K. (2022), "Spirituality in Serious Illness and Health," JAMA, 328 (2). DOI: 10.1001/jama.2022.11086.

Thank you for providing us with recent studies so relevant to the subject of our research. We add the revision you propose to give more background to our justification.

The authors mention in the discussion that more research on psychosocial aspects of R/S should be made to determine the pathways through which R/S acts on health. The sentence sounds like the authors think that enough research on psychosocial factors in R/S would be able to explain all R/S effects on health(around L883). Albeit it is definitely true that more such research is needed, the authors may consider including a line or two on the possibility that R/S works on health through yet unknown paths that could include some form of divine interventions. While this is surely a bold statement, I believe that it from a scientific theory perspective is a possibility that researchers in all fields should keep open.

We agree with you. Despite all the progress of this branch of knowledge, there is still the possibility that there are unknown pathways by which R/S can affect health. We have added this suggestion by adapting it to the text in lines 479-484. Thank you for pointing out this possibility that allows us to open a path towards interesting avenues of future research.

Specific comments

L173: the aim now has a spelling error… delete “may” Done, thank you.

L240: I believe “search string” is more commonly used than “equation”, but I might be wrong? Done, thank you.

L978: a recently published protocol of a cohort study includes cancer patients in a longitudinal setup to study R/S and health. You may consider referencing it as an example of a study and design that might mitigate the shortcomings of cross-sectional studies on R/S and health in cancer patients.

Stripp, T. K., Wehberg, S., Büssing, A., Andersen-Ranberg, K., Jensen, L. H., Henriksen, F. L., Laursen, C. B., Søndergaard, J., and Hvidt, N. C. (2022), "Protocol for EXICODE: the EXIstential health COhort DEnmark—a register and survey study of adult Danes," BMJ Open, 12 (6), e058257. DOI: 10.1136/bmjopen-2021-058257.

Thank you very much again for your input. We believe that this study is perfect to exemplify the need to develop longitudinal designs that will help us reduce the problems associated with cross-sectional designs in studies that relate R/S and health in populations with serious diseases, such as cancer. We have added a sentence alluding to this idea, quoting the study you propose (lines 568-572).

Appendix A:

For brevity, you may note only e.g. “52% female” and omit the percentage for males. The reader will be able to figure out that the rest (48%) are males – for all practical reasons anyway… The number of transsex/gender persons is usually very low in studies like these.

We agree with the suggestion. We have eliminated the percentage of men in each study, for brevity and to facilitate the interpretation of the table.

Island should be Iceland. Done, thank you.

Again, thank you very much for your general and specific comments. We believe that all of them have greatly improved the quality of the manuscript. We believe that after all the revisions we have obtained as a result an interesting contribution to the scientific community, with important implications for different fields of knowledge.
